# Near-Field-to-Far-Field RCS Prediction Using Only Amplitude Estimation Technique Based on State Space Method

**Jinhai Huang** [1,2] , **Jianjiang Zhou** [1,*] **and Yao Deng** [1]

1   Key Laboratory of Radar Imaging and Microwave Photonics, Nanjing University of Aeronautics and
    Astronautics, Nanjing 210016, China; jinhaiwell@nuaa.edu.cn (J.H.); dy111896@nuaa.edu.cn (Y.D.)
2   College of Electronic Engineering, Guilin Institute of Information Technology, Guilin 541004, China
*   Correspondence: zjjee@nuaa.edu.cn

**Abstract:** Measuring the radar cross-section (RCS) of a far-field (FF) target in engineering can be challenging, especially when remote measurement is difficult. To overcome this challenge, an FF RCS can be predicted by near-field (NF)-extrapolated transformation. However, due to the relative error between the theoretical and measured electric field (E-field) values in a NF, the extrapolation calculation of a FF can be carried out by correcting the NF amplitude. This paper proposes the use of the state space method (SSM) to estimate the amplitude-only of NF E-fields for improving the prediction accuracy of FFs. The simulation results demonstrate that the SSM can estimate NF amplitude, which can be transformed into a FF, and which can lead to improved prediction accuracy when compared to reference-FF-calculated and to circular-NF-to-FF-transform-(CNFFFT)-calculated RCSs.

**Keywords:** radar cross section (RCS); state space method (SSM); near-field-to-far-field transformation (NFFFT); circular-near-field-to-far-field transform (CNFFFT)

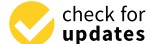



## 1. Introduction

### 1.1. Background and Motivation

Direct measurements of the RCS of electrically electric large objects and of the radiation pattern of large antennas is frequently very challenging, because a RCS requires a target object to be lit by an incident wave that is considered to be a plane wave. The relationship $R \geq 2D^2/\lambda$, where $D$ is the target's real maximum size and where $\lambda$ is the operating electromagnetic (EM) wavelength, is used to determine this distance in the FF. Assuming the target size is D = 10 m and the wavelength is $\lambda$ = 0.03 m, the FF measurement distance needs to satisfy the criterion of 6666.67 m. In such conditions, constructing a test environment up to the kilometer scale is considerably difficult, especially for large-size targets. Therefore, predicting a FF RCS from an NF extrapolation is an important research area. Recent studies have demonstrated that by processing the scattered EM field detected by a scanning probe around the target in the NF region, NF-to-FF transformation [1–4] (NFFFT) technology can predict the target's RCS in other regions.

The scattering behavior of EM waves is dependent on frequency, and higher-frequency measurements typically require the use of full-scale structures instead of scale models. Consequently, measuring medium-sized parts, let alone full-scale structures, necessitates outdoor radar ranging. The characterization of an object using scattered radiation is commonly referred to as a backscattering problem, and numerous solution methods can be found in the literature. The choice of a particular solution method depends on the features of the object that are relevant to the application. In a previous study [5], the coherent Doppler tomography (CDT) method was utilized, to determine FF RCSs from NF measurements. This paper highlights the challenges associated with measuring the backscattering properties of large specimens using EM waves, and it emphasizes the need for full-scale structure measurements, due to the frequency-dependent scattering behavior of EM waves. However, it is important to note that any measurement system is subject

to both theoretical and practical errors, and the paper does not address the challenges associated with NF inversion of FF RCSs.

In the paper [6], the circular-NF-to-FF transformation (CNFFFT) method is proposed by the authors as a technique to predict FF RCSs, using NF measurements collected on a circular path around a target. However, the CNFFFT algorithm requires measurements over the full 360-degree range, which may not always be feasible. To address this issue, the authors consider the CNFFFT algorithm as an azimuthal filtering process, and they develop a formula capable of transforming data measured over a partial rotation. The paper provides guidelines for the NF data support required to achieve the desired accuracy in CNFFFT results below 360 degrees. The numerical simulations presented in the paper demonstrate that the results of this partial rotation formulation are consistent with the full-circle CNFFT results from previous studies.

### 1.2. Primary Contribution

The significant contributions of this article are as follows:

(1) From a fundamental perspective, the target's incident and reception functions in the NF testing environment are derived, using the dyadic Green's function. This resolves the theoretical derivation of the received signal in complex EM environments.

(2) By considering amplitude as a crucial intensity characteristic for predicting FFs, the amplitude feature extraction of the NF signal is achieved, using the SSM. This addresses the coefficient calculation for near-field-to-far-field transformations.

(3) Based on the solution approach of CNFFFT, the near-field-to-far-field transformation kernel is derived and improved. This effectively resolves the prediction of RCSs from the NF transformation to the FF.

In this paper, a novel approach to amplitude-only estimation in the NFFFT process is introduced, which is based on the SSM. The proposed method is compared to the CNFFFT algorithm. The theoretical framework is presented in Section 2, followed by the experimental setup and analysis results in Section 3. Finally, the conclusion is provided in Section 4.

### 2. Methods

RCS is a critical metric used to measure the amount of radar energy scattered by an object, and it is essential in determining the detectability of the object. To achieve precise RCS determination, radar data collection plays a crucial role. Monostatic radar measurements offer the means to acquire NF RCS measurements, facilitating the prediction of RCS values for scattering objects with unknown properties. This prediction is reliant upon utilizing the obtained NF RCS measurements as a foundational reference. Nonetheless, achieving precise and reliable results necessitates the deployment of a suitable conversion algorithm that considers crucial factors, including antenna direction and the arbitrary measurement location of the target. Furthermore, it is vital to ensure the fulfillment of sampling requirements, to maintain the accuracy of RCS measurements. In a study conducted by the authors [7], a method was proposed for accurately modeling the behavior of NF scattering, which is utilized in this study. The method involves a radiation reflector model and a multilevel-plane-wave decomposition approach. By decomposing the incident waves into multiple plane waves with varying frequencies and velocities, and considering the scattering waves as emanating from a set of reflectors, the intricate relationship of NF scattering is effectively captured. The proposed method provides a comprehensive framework for analyzing and comprehending the complex interactions between incident waves and scattering objects in a NF region.

The model to this method comprises four key aspects:

(1) The representation of the object involves a hierarchical structure of recognized or anticipated shapes of scattering centers.

(2) A model is established to illustrate the relationship between the incident wave and the scattered wave.

(3) The computation of the RCS necessitates solving a set of linear equations.
(4) Based on the results obtained from step (3), the RCS of the object is subsequently calculated.

### 2.1. NF Linearized Scattering Model

The NF structure of the antenna under test (AUT) in relation to an unidentified test object surface is depicted in Figure 1. Within this configuration, each scattering point is associated with its specific propagation vector $k_i$, which is oriented towards the center of the corresponding facets. This implies that each scattering point can be treated as an individual incident field $E^i$. The spatial domain in this study is represented by the three-dimensional (3D) coordinates $(x, y, z)$ with the distance vector $\boldsymbol{r}$ indicating the separation between the scattering center point and the transmitting point. The distance vector of scattering center and the $i$-th scattering point can be written as

$$\boldsymbol{r} = x\hat{e}_x + y\hat{e}_y + z\hat{e}_z \tag{1}$$

$$\boldsymbol{r}_i = x_i\hat{e}_x + y_i\hat{e}_y + z_i\hat{e}_z \tag{2}$$

where $(\hat{e}_x, \hat{e}_y, \hat{e}_z)$ is the unit vector along itself direction.

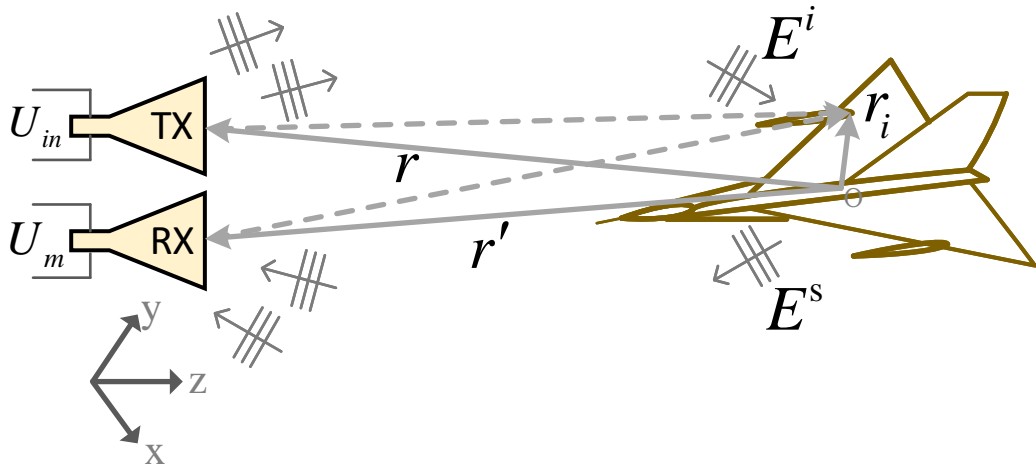

**Figure 1.** The design of the antenna configuration used for NF testing of the target.

The general bistatic scenario, as depicted in Figure 1, provides the basis for modeling the linearized forward operator equations. These equations can then be adapted for the monostatic case. In this setup, the equivalent current density distribution $J_s$ in free space corresponds to the radar antenna operating in transmit mode. It has been normalized concerning the excitation voltage $\boldsymbol{U}_{in}$ at the antenna feed port. This normalization process results in the following equation, as documented in [7].

$$\boldsymbol{J}_s(\boldsymbol{r}) = \boldsymbol{U}_{in}(\boldsymbol{r})\mathrm{w}_T(\boldsymbol{r}) \tag{3}$$

where $\mathrm{w}_T$ represents an transmitted impedance weighting formula determined by the reference coordinate system.

The incident field $\boldsymbol{E}^i(\boldsymbol{r})$ at the scatterer, induced by the current distribution, can be expressed using the dyadic Green's function $\overline{\boldsymbol{Gn}}(\boldsymbol{r}_i, \boldsymbol{r})$ of free space, as shown in the following formula:

$$\boldsymbol{E}^i(\boldsymbol{r}) = -\frac{jkZ_F}{4\pi} \iiint_{V_T} \overline{\boldsymbol{Gn}}(\boldsymbol{r}_i, \boldsymbol{r}) \boldsymbol{J}_s(\boldsymbol{r}) d^3\boldsymbol{r} \tag{4}$$

$$\overline{\boldsymbol{Gn}}(\boldsymbol{r}_i, \boldsymbol{r}) = \left(\bar{\mathbf{I}} + \frac{1}{k^2}\nabla\nabla\right) \frac{\exp(-jk\|\boldsymbol{r}_i - \boldsymbol{r}\|)}{\|\boldsymbol{r}_i - \boldsymbol{r}\|} \tag{5}$$

Herein, the symbol $V_T$ in Equation (4) is determined by the spatial extent of the scatterer during transmission. The symbol $Z_F$ represents the characteristic impedance of free space. The scalar wave number $k$ in equation (4) is the absolute value of the wave number vector $\boldsymbol{k}$, where $\boldsymbol{k}$ is defined as $\boldsymbol{k} = k_x \hat{e}_x + k_y \hat{e}_y + k_z \hat{e}_z$ in Equation (5). The magnitude of $\boldsymbol{k}$, denoted as $|\boldsymbol{k}|$, is calculated as $\sqrt{k_x^2 + k_y^2 + k_z^2}$. The operator $\nabla$ represents the gradient operator, and $\bar{\bar{\mathbf{I}}}$ is the unit dyad. Additionally, $\|\boldsymbol{r}_i - \boldsymbol{r}\|$ denotes the norm of distance vector between the target and the transmitting antenna under test (AUT). To simplify the computation, the exponential term in Equation (5) can be evaluated using the provided Equation [8].

$$\frac{\exp(-jk\|\boldsymbol{r}_i - \boldsymbol{r}\|)}{\|\boldsymbol{r}_i - \boldsymbol{r}\|} = \int\int_{-\infty}^{+\infty} \frac{-j}{2\pi k_z} e^{-jk_z|z_i - z|} e^{-jk_x(x_i - x)} e^{-jk_y(y_i - y)} dk_x dk_y \qquad (6)$$

where

$$k_z = \begin{cases} \sqrt{k^2 - k_x^2 - k_y^2}, & for \ k^2 > k_x^2 + k_y^2 \\ -j\sqrt{k_x^2 + k_y^2 - k^2}, & for \ k^2 < k_x^2 + k_y^2 \end{cases} \qquad (7)$$

In a similar manner, the dyadic Green's function of a receiving field in free space can be expressed as follows.

$$\overline{\boldsymbol{Gn}}'(\boldsymbol{r}_i, \boldsymbol{r}') = \left(\bar{\bar{\mathbf{I}}} + \frac{1}{k^2}\nabla\nabla\right)\frac{\exp(-jk\|\boldsymbol{r}_i - \boldsymbol{r}'\|)}{\|\boldsymbol{r}_i - \boldsymbol{r}'\|} \qquad (8)$$

where the 3D distance vector in space between the source point and the receiving antenna is denoted by $\boldsymbol{r}'$. The NF scattering signal received by the antenna can be obtained by the following expression.

$$\boldsymbol{E}^s(\boldsymbol{r}') = \iiint_V \alpha_i \overline{\boldsymbol{Gn}}'(\boldsymbol{r}_i, \boldsymbol{r}') \boldsymbol{E}^i(\boldsymbol{r}) \bar{\kappa}(\boldsymbol{r}') d^3\boldsymbol{r}' \qquad (9)$$

where V in Equation (9) is determined by the spatial extent of the scatterer, the coefficient $\alpha_i$ represents the magnitude of the $i$-th component, the polarization phase term $\bar{\kappa}(\boldsymbol{r}') = \exp(j\boldsymbol{k} \cdot \boldsymbol{r}')$ is utilized to relocate the phase reference center of the target scattering function from the radar receiving antenna to the center of the target. By utilizing this term, the phase alignment can be adjusted, thereby facilitating the accurate localization of the target scattering characteristics. The polarization phase term plays a crucial role in ensuring that the phase reference center of the target scattering function is properly aligned with the radar receiving antenna. This alignment is essential for achieving a precise and reliable localization of the target's scattering properties. The phase adjustment provided by this term allows for a more accurate analysis and interpretation of the scattering behavior, leading to enhanced understanding and characterization of the target's EM response.

The voltage measured $U_m$ at the receiving antenna can be expressed as a function of the scattering field and the weighting function $\mathrm{w}_R(\boldsymbol{r}')$ of the receiving impedance, based on the Born approximation [9] and reciprocity, resulting in

$$U_m = \iiint_{V_R} \mathrm{w}_R(\boldsymbol{r}') \boldsymbol{E}^s(\boldsymbol{r}') d^3\boldsymbol{r}' \qquad (10)$$

where $V_R$ in Equation (10) is determined by considering the spatial extent of the scatterer as observed by the receiving system.

To optimize the error between the actual measured voltage and the theoretically derived receiving field voltage, the amplitude $\alpha_i$ correction can be estimated using an objective function constructed from the E-field values, which will be discussed in the Section 2.2.

### 2.2. Estimation of Magnitude

The auto-regressive moving average model [10,11], which has both auto-regressive and moving average structures can be stated as follows in terms of linear systems and control theory:

$$
\begin{aligned}
E^i(n+1) &= AE^i(n) \\
E^s(n) &= \Psi E^i(n) + \Theta \epsilon(n)
\end{aligned}
\tag{11}
$$

where the $n$-th component of the $\boldsymbol{E}^i(\boldsymbol{r})$ and $\boldsymbol{E}^s(\boldsymbol{r}')$, respectively, is $E^i(n)$ and $E^s(n)$. $A \in C^{M \times M}$ denotes the open-loop matrix, $\Psi \in C^{M \times 1}$ and $\Theta \in C^{1 \times M}$ are the coefficient of regressors, white noise with no time dependence and a mean of zero is represented by the function $\epsilon(n)$.

To compute the triplet $(A, \Psi, \Theta)$, the SSM is used to construct a Hankel matrix **H** based on N measurements. It can be expressed as

$$
\mathbf{H} = \begin{bmatrix}
\frac{E^s(1)}{U_m} & \frac{E^s(2)}{U_m} & \cdots & \frac{E^s(L)}{U_m} \\
\frac{E^s(2)}{U_m} & \frac{E^s(3)}{U_m} & \cdots & \frac{E^s(L+1)}{U_m} \\
\vdots & \vdots & \vdots & \vdots \\
\frac{E^s(N-L+1)}{U_m} & \frac{E^s(N-L+2)}{U_m} & \cdots & \frac{E^s(N)}{U_m}
\end{bmatrix}
\tag{12}
$$

where $\frac{E^s(N)}{U_m}$ represents the normalization under the same measured voltage, the component $E^s(N)$ represents a specific portion of the field $E^s(r')$, the value in brackets corresponds to the largest integer that is less than or equal to the provided value. The length of the correlation window is denoted as $L$, and in order to determine the value of $N$, it is heuristically set to be equal to half of the window length, i.e., $N = [L/2]$. Subspace decomposition techniques utilize the eigen-structure of Hankel matrices to estimate the parameters of linear time-invariant(LTI) signal models [12].

To extract the relevant signal components, the application of the singular value decomposition (SVD) to the matrix **H** leads to a decomposition of the form $\tilde{\mathbf{H}} = \mathbf{U}_{sn}\mathbf{\Sigma}_{sn}\mathbf{V}_{sn}^*$, where the subscript '$sn$' to denote the signal component corrupted by noise, the left-unitary matrix $\mathbf{U}_{sn}$ and the conjugate transpose of the right-unitary matrix $\mathbf{V}_{sn}^*$ are orthogonal matrices, and $\mathbf{\Sigma}_{sn}$ is a diagonal matrix with singular values on the diagonal. The SVD allows us to analyze the contributions of each singular value to the overall signal representation and identify the dominant signal components.

Furthermore, the matrix **H** can be further factorized by employing the balanced coordinate transformation, as described in the work by reference [13]

$$
\tilde{\mathbf{H}} = \tilde{\mathbf{\Omega}}\tilde{\mathbf{\Gamma}}
\tag{13}
$$

where the finite-rank observability matrix is denoted as $\tilde{\mathbf{\Omega}} = \mathbf{U}_{sn}\mathbf{\Sigma}_{sn}^{1/2}$, and the controllability matrix is represented by $\tilde{\mathbf{\Gamma}} = \mathbf{\Sigma}_{sn}^{1/2}\mathbf{V}_{sn}^*$. This transformation allows for a more refined decomposition of **H** into its constituent parts, providing a more comprehensive understanding of the underlying signal structure. By applying the balanced coordinate transformation, the matrix **H** can be expressed as a product of matrices that capture the specific characteristics and relationships within the signal data. This factorization technique enhances the interpretability and analytical capabilities of the signal processing algorithm.

Additionally, the open-loop matrix **A** can be derived by employing the observability matrix $\tilde{\mathbf{\Omega}}$, which captures the relationships of the system.

$$
\mathbf{A} = \left(\tilde{\mathbf{\Omega}}_{-r\ell}^*\tilde{\mathbf{\Omega}}_{-r\ell}\right)^{-1}\tilde{\mathbf{\Omega}}_{-r\ell}^*\tilde{\mathbf{\Omega}}_{-r1}
\tag{14}
$$

where the matrices $\tilde{\mathbf{\Omega}}_{-r\ell}$ and $\tilde{\mathbf{\Omega}}_{-r1}$ are modified versions of matrix $\tilde{\mathbf{\Omega}}$, with the final and first rows removed, respectively. By utilizing the observability matrix, the necessary

information about the system's observability can be extracted, enabling the calculation of the open-loop matrix **A**. Furthermore, the eigenvalues of matrix **A** can be computed as

$$\lambda\{\mathbf{A}\} = \{\lambda_1, \lambda_2, \cdots, \lambda_M\} \tag{15}$$

In engineering applications that involve linear transformations or LTI systems, a sequence of input vectors corresponds to a sequence of output vectors. Each input vector is considered as the input to an LTI system, and the associated eigenvalue can be interpreted as the amplification factor of the linear system input. By considering the eigenvalue as its gain, the estimation of signal amplitude can be achieved, and this can be accomplished by employing Equation (14).

$$\alpha_i = -\frac{\log|\lambda_i|}{\Delta\varphi} \tag{16}$$

where $\Delta\varphi$ represents the angular difference between the azimuth of the target and the increment angle of EM wave radiation.

*2.3. Near-Field-to-Far-Field Transformation*

To obtain scattering data in the FFs, complex processing is necessary for the radiation field at each angle. Suppose the FF distance vector is denoted as $\mathbf{r}_{FF}$, and the convolution kernel function $\mathrm{K}(\mathbf{r}_{FF})$ of the FF transform is convolved with the NF scattering signal for all transmission and reception angles. This convolution process yields the scattering signal of the FF E-field, which is expressed by [14].

$$\mathbf{E}^s(\mathbf{r}_{FF}) = \mathbf{E}^s(\mathbf{r}') * \mathrm{K}(\mathbf{r}_{FF}) \tag{17}$$

where the interaction between the incident plane wave and the scattering object in the FF region is described by an equation that involves the convolution operation, represented by the asterisk symbol. This equation captures the scattering phenomena and allows for the analysis of the scattered field in the FF region. Mathematically, it can be expressed as

$$\mathbf{E}^s(\mathbf{r}_{FF}) = \iiint\limits_{V} \alpha_i \overline{\mathbf{Gn}}(\mathbf{r}_{FF_i}, \mathbf{r}_{FF}) \mathbf{E}^i(\mathbf{r}_{FF}) \bar{\kappa}(\mathbf{r}_{FF}) d^3\mathbf{r}_{FF} \tag{18}$$

where V in (18) represents the integration volume of FF target, the vector $\mathbf{r}_{FFi}$ represents the $i$-th coordinate vector in the FF region, $\bar{\kappa}(\mathbf{r}_{FF}) = \exp(-j\mathbf{k} \cdot \mathbf{r}_{FF})$ represents the FF polarization phase term, the magnitude $\alpha_i$ can be obtained by performing the inverse Fourier Transform (IFT) on Equation (9), resulting in

$$\alpha_i = \frac{F^{-1}[\mathbf{E}^s(\mathbf{r}')]}{\overline{\mathbf{Gn}}'(\mathbf{r}_i, \mathbf{r}')\mathbf{E}^i(\mathbf{r})} \tag{19}$$

The incident E-field in the FF region for the target can be acquired using the FF dyadic Green function $\overline{\mathbf{Gn}}(\mathbf{r}_{FF_i}, \mathbf{r}_{FF})$, serving as

$$\mathbf{E}^i(\mathbf{r}_{FF}) = -\frac{jkZ_F}{4\pi} \iiint\limits_{V_T} \overline{\mathbf{Gn}}(\mathbf{r}_{FF_i}, \mathbf{r}_{FF}) \mathbf{J}_s(\mathbf{r}_{FF}) d^3\mathbf{r}_{FF} \tag{20}$$

Next, substitute Equations (19), (20) into Equation (18), we have

$$
\begin{aligned}
\boldsymbol{E}^s(\boldsymbol{r}_{FF}) &= F\left[\frac{F^{-1}[\boldsymbol{E}^s(\boldsymbol{r}')]}{\overline{\boldsymbol{Gn}}'(\boldsymbol{r}_i,\boldsymbol{r}')\boldsymbol{E}^i(\boldsymbol{r})}\overline{\boldsymbol{Gn}}(\boldsymbol{r}_{FF_i},\boldsymbol{r}_{FF})\boldsymbol{E}^i(\boldsymbol{r}_{FF})\right] \\
&= F\left[\frac{\overline{\boldsymbol{Gn}}(\boldsymbol{r}_{FF_i},\boldsymbol{r}_{FF})\boldsymbol{E}^i(\boldsymbol{r}_{FF})}{\overline{\boldsymbol{Gn}}'(\boldsymbol{r}_i,\boldsymbol{r}')\boldsymbol{E}^i(\boldsymbol{r})}F^{-1}\left[\boldsymbol{E}^s(\boldsymbol{r}')\right]\right] \\
&= F\left[\omega_i F^{-1}\left[\boldsymbol{E}^s(\boldsymbol{r}')\right]\right]
\end{aligned}
\tag{21}
$$

where the Fourier Transform (FT) is denoted by the symbol $F[\cdot]$. In this case, $\omega_i$ represents the NF-to-FF ratio, and the relationship between them can be formulated as follows:

$$
\omega_i = \frac{\overline{\boldsymbol{Gn}}(\boldsymbol{r}_{FF_i},\boldsymbol{r}_{FF})\boldsymbol{E}^i(\boldsymbol{r}_{FF})}{\overline{\boldsymbol{Gn}}'(\boldsymbol{r}_i,\boldsymbol{r}')\boldsymbol{E}^i(\boldsymbol{r})}
\tag{22}
$$

Subsequently, the convolution kernel function described in Equation (17) can be expressed as

$$
\mathrm{K}(\boldsymbol{r}_{FF}) = F[\omega_i]
\tag{23}
$$

### 3. Simulation, Analysis and Calibration Discussion

To assess the rationality of the proposed method based on the existing research foundation [15,16], a target model for FEKO simulation is established, as depicted in Figure 2, with a frequency of 2 GHz. The simulation process is configured within a monostatic environment, where equal integration volumes are employed for transmission and reception, regardless of whether it is in the NF or FF conditions. Specifically, in both the NF and FF, $V_T = V_R$. Using the method proposed in this paper, the following three experiments are conducted to compute the NF and FF results through simulation and analysis.

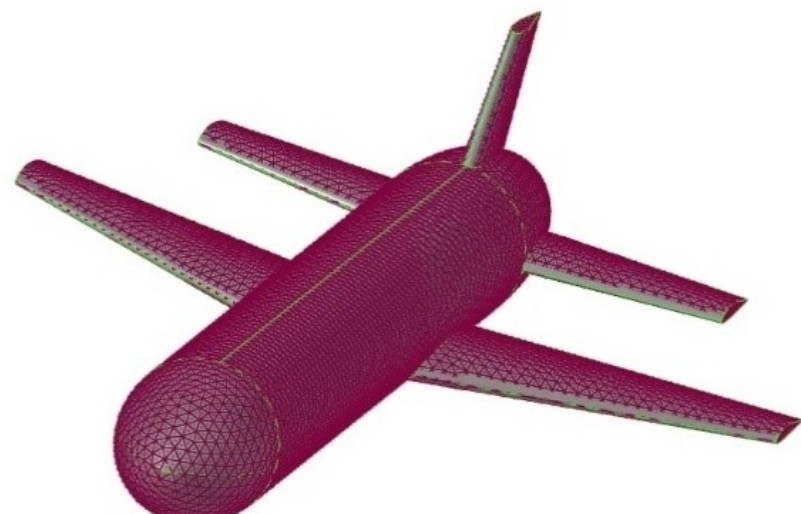

**Figure 2.** The geometry model of test target.

(1) The E-fields and RCS values are computed at different distances in the NF, which the magnitudes are extracted based on the proposed SSM estimation method, and the E-fields and RCS values in the FF are derived.

(2) The RCS values transitioning from the NFs to the FF are calculated, along with the E-fields and RCS values simulated independently in the FF. A comparison is made between the RCS corrected with the NF magnitude extracted by SSM and the NFFFT predicted RCS values without extracting the NF magnitude, and the error is calculated.

(3)    The E-fields and RCS values simulated independently in the FF serve as reference data for comparison. The NFFFT corrected with the NF magnitude extraction and the classic CNFFFT calculations are contrasted.

### 3.1. Experiment 1

At the position of the target point source depicted in Figure 1, the complex model phenomenon of multiple zeros and peaks in the monostatic reflectivity arrangement of point scatterers necessitates the sampling of the complex signal of the target response. This ensures that each point scatterer is associated with a randomly generated complex reflectance coefficient following a normal distribution. To obtain experimental results of RCS at different distances in the NF, it is necessary to construct the NF scattering data of the target. The target size is set to 2 m, and the frequency is set to 2GHz, consistent with the FEKO configuration, based on the proposed NF model theory method in this paper. The distances are defined as NF1 = 1 m, NF2 = 10 m, NF3 = 25 m, and NF4 = 50 m. Multiple scattering centers can be employed for integration and summation to acquire the NF E-field scattering data for the target. The RCS values are then calculated based on the NF computation formula as follows:

$$\sigma_{NF} = \lim_{r \to \infty} 4\pi r^2 \frac{\left|\boldsymbol{E}^s\left(\boldsymbol{r}'\right)\right|^2}{\left|\boldsymbol{E}^i\left(\boldsymbol{r}\right)\right|^2} \tag{24}$$

The NF scattering E-fields and RCSs are simulated for four distinct scattering distances, taking into account the HH and VV polarization modes. The distances, denoted as NF1, NF2, NF3, and NF4, are arranged in increasing order from short to long. Figure 3 illustrates the NF scattering E-fields, while Figure 4 displays the corresponding RCSs. It is evident from the figures that as the distance increases, the values of the E-fields decrease, and the RCSs exhibit corresponding variations.

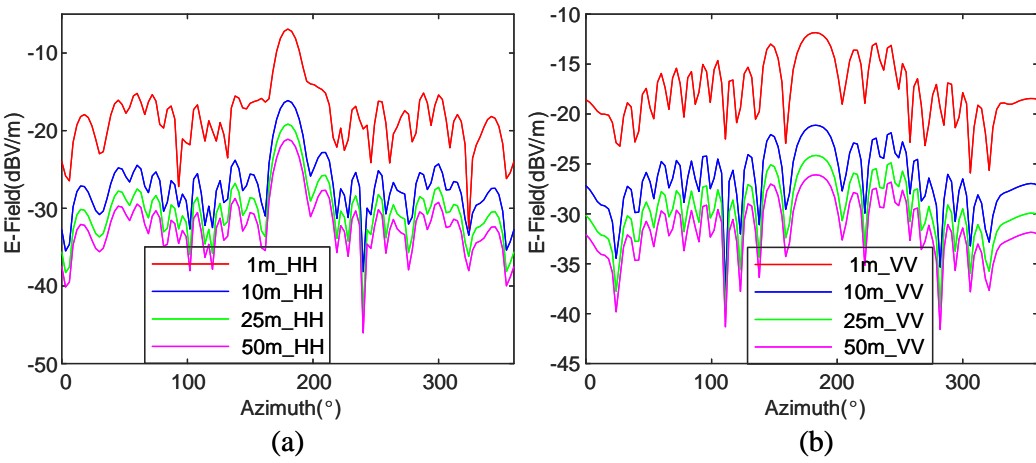

**Figure 3.** (**a**) NF E-Field in HH polarization mode; (**b**) NF E-Field in VV polarization mode.

Based on the theory presented in Section 2, the NF amplitude is extracted using the SSM method to achieve optimal FF transformation. The construction of the Hankel function relies on the NF scattering field at the corresponding distance. For computational simplicity, the voltage parameter in Equation (10) can be replaced with the E-field. The numerator of the Hankel function is computed using FEKO, while the denominator corresponds to actual measurement values. The main SVD extraction process enables accurate estimation of the amplitude.

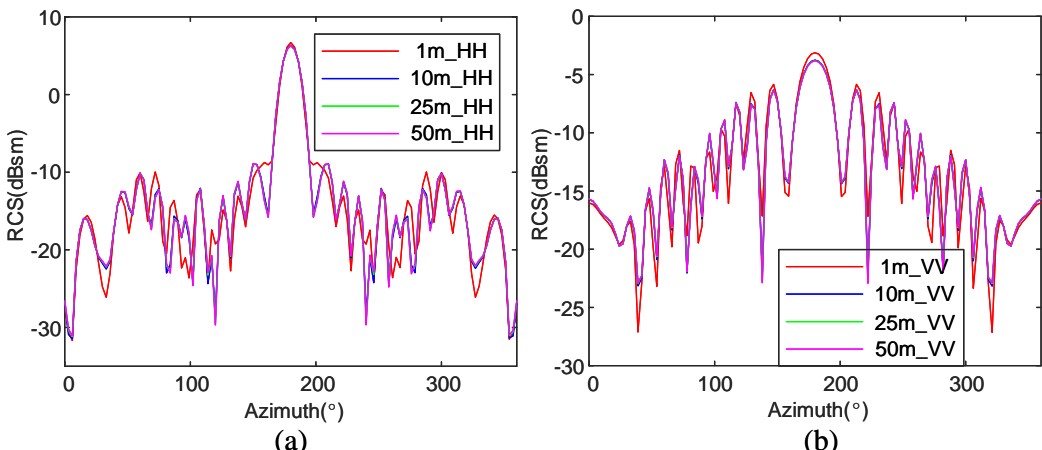

**Figure 4.** (**a**) NF RCS in HH polarization mode; (**b**) NF RCS in VV polarization mode.

To perform the near-field-to-far-field transformation, the kernel specified in Equation (23) is employed to convert the NF E-field into the FF RCS. The FF input voltage $U_{in}(r_{FF})$ is approximated by the NF excitation voltage $U_{in}(r)$, and $E^i(r)$ is obtained using Equation (4). Subsequently, the kernel is derived by substituting the distance of the NF into the dyadic Green function presented in Equations (5) and (8), followed by applying the FT. Finally, the FF RCS can be computed by substituting the near-field-to-far-field kernel into Equation (17), leading to

$$\sigma_{FF} = 4\pi \frac{\left|E^s(r_{FF})\right|^2}{\left|E^i(r_{FF})\right|^2} \tag{25}$$

Figures 5 and 6 present the FF E-field and RCS, respectively. In these figures, the NF simulated data are depicted by the red line, while the FF results obtained using FEKO are represented by the black line. The blue line in both figures corresponds to the results of the near-field-to-far-field transformation, which the FF distance in the transformation simulation procedure is set to 1000 m. Notably, the near-field-to-far-field transformation exhibits superior accuracy compared to the FF RCSs, not only in the HH polarization mode but also in the VV mode.

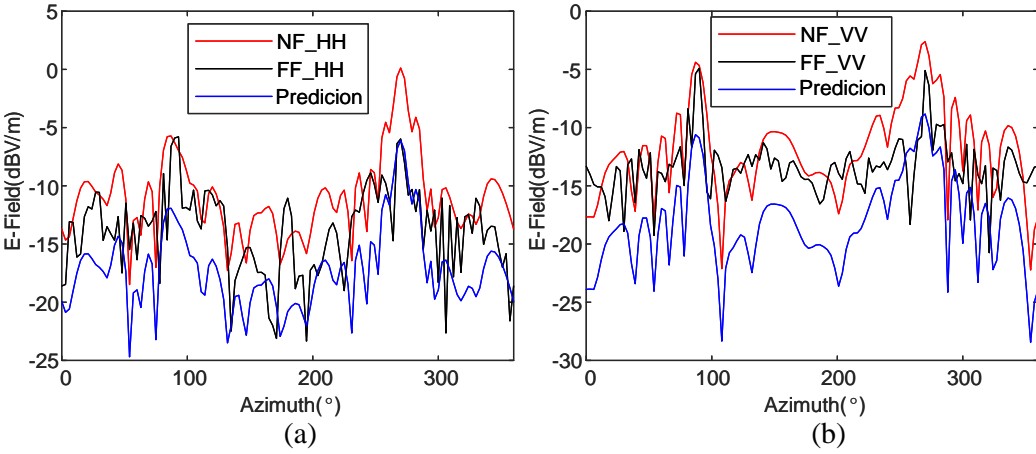

**Figure 5.** The E-Field comparison of the NF, FF, and near-field-to-far-field transformations: (**a**) HH polarization mode; (**b**) VV polarization mode.

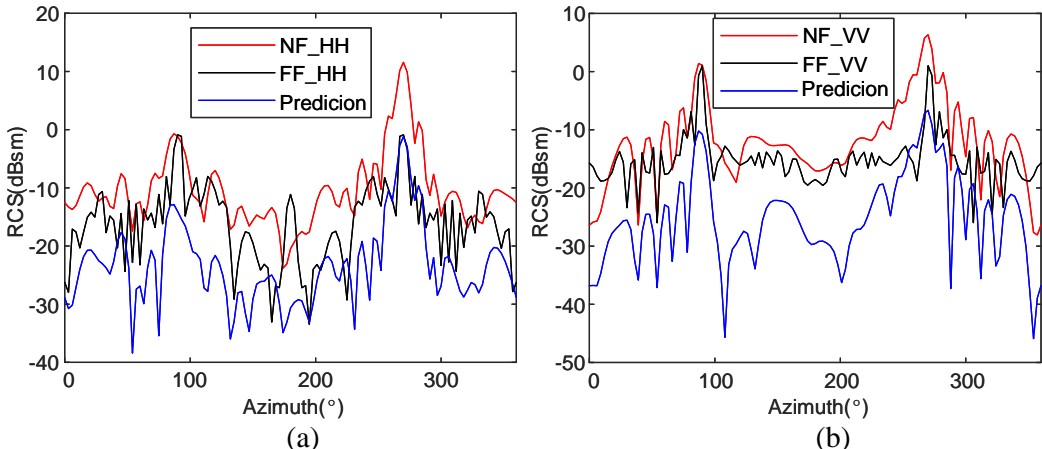

**Figure 6.** The RCS comparison of the NF, FF, and near-field-to-far-field transformations: (**a**) HH polarization mode; (**b**) VV polarization mode.

*3.2. Experiment 2*

In Section 3.1, the near-field-to-far-field transformations are simulated by the principles outlined in this paper. To accurately represent the NF characteristics of the target, the NF data transformed into the FF domain are simulated using FEKO software. The simulation involved configuring the relevant NF parameters, including three distances NF1 = 1 m, NF2 = 10 m, and NF3 = 25 m.

Furthermore, the predicted results obtained from Section 3.1 are refined by incorporating the amplitude estimation derived from the SSM method. This additional step aimed to improve the accuracy of the predictions. To evaluate the effectiveness of the proposed approach, a comprehensive analysis is conducted by comparing the errors between the theoretical simulations and the FEKO simulations.

In order to observe the impact of the NF-to-FF transformation, the FF results are obtained in both HH and VV polarization modes. Figure 7 illustrates the simulation results, showcasing different data sets. The NF1 data are represented by the red lines, the NF2 results are depicted by the black lines, and the NF3 results are also displayed using green dotted lines. Additionally, the blue lines represent the predicted results obtained through the simulation using FEKO. The comparison of these results allows for an assessment of the accuracy and efficacy of the proposed approach.

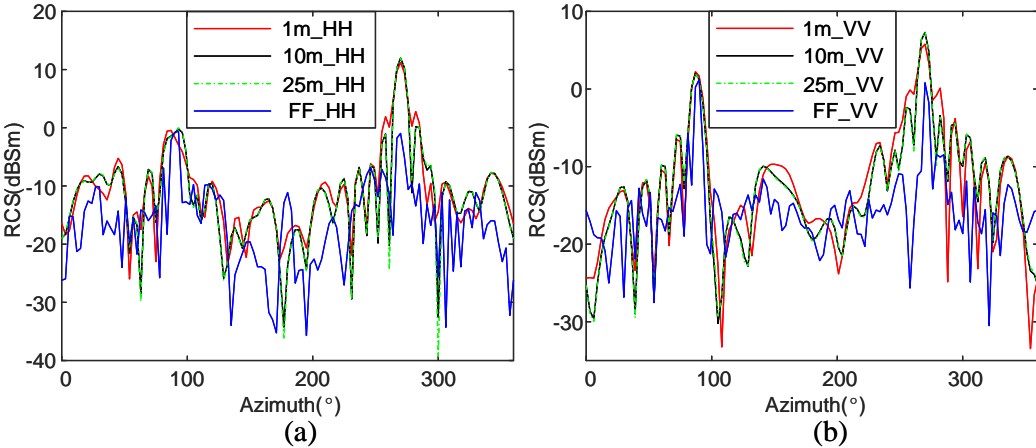

**Figure 7.** The NF data and FF data were simulated using FEKO: (**a**) HH mode; (**b**) VV mode.

The analysis of the figures depicted in Figures 3–6 leads to the conclusion that as the distance separating the target and the measurement device increases, the NF RCSs exhibit a closer resemblance to the FF RCSs. This observation is attributed to the relationship

between the square of the NF distance and the NF scattering field, which allows for the approximation of the NF E-field and RCS to the FF RCS.

To distinguish between the results obtained from NFFFT and the simulation performed in FEKO, the NF amplitude estimated by SSM is utilized to adjust the NF data simulated in FEKO. As a result, corrected values for the FF reference are obtained. A comparison is subsequently made between these corrected values and the data predicted by NFFFT, as derived in this paper. The error is calculated to assess the advantages of the proposed method in this study.

$$Err = 10 \log 10 \left( \|\sigma_1 - \sigma_2\|_2^2 \right) \quad (26)$$

A strong correlation is evident between the RCS acquired from the near-field-to-far-field transformation and the amplitude correction line, as illustrated in Figures 8 and 9 for various polarization modes in relation to the E-field. The effectiveness of the SSM method in estimating amplitudes is confirmed by conducting error calculations, which establish a high level of agreement with the theoretical framework put forth in this investigation.

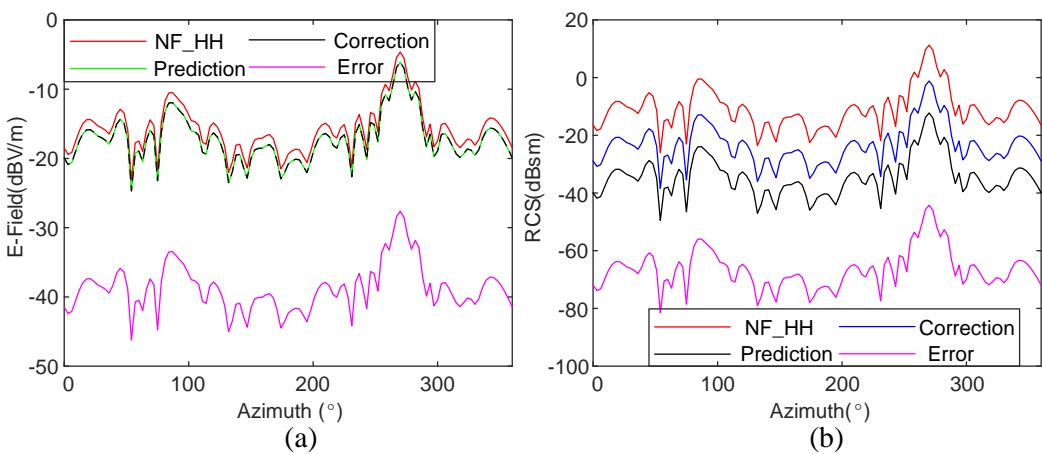

**Figure 8.** The comparison of the NF, near-field-to-far-field transformations prediction, FF reference correction and error in HH mode: (**a**) E-Field; (**b**) RCS.

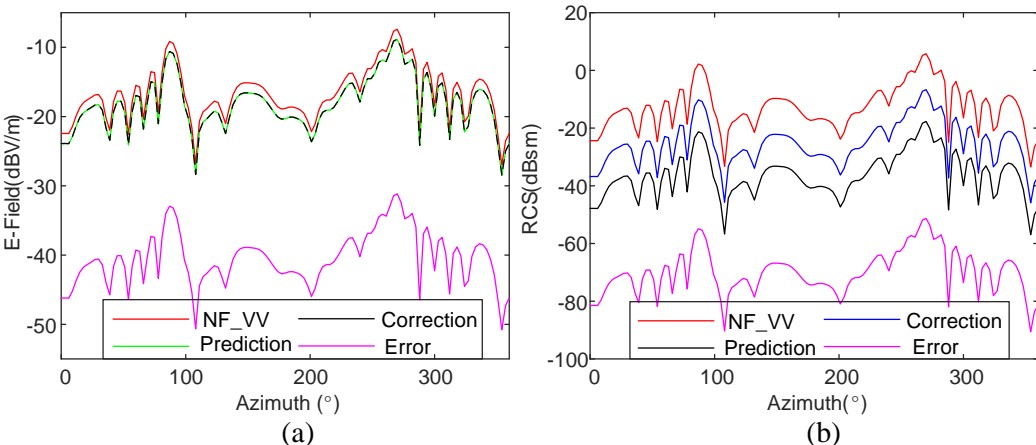

**Figure 9.** The comparison of the NF, near-field-to-far-field transformations prediction, FF reference correction and error in VV mode: (**a**) E-Field; (**b**) RCS.

### 3.3. Experiment 3

To further validate the effectiveness of the method proposed in this paper, the CNFFFT method is introduced for comparison under the same parameter configuration. According to the introduction of CNFFFT in reference [2], the algorithm filters the data based on the rotation angle around the target along a circular path. It predicts the FF RCS using the

collected NF measurement results and also improves the ability to transform data that are not measured within a complete 360-degree rotation. This is particularly useful in scenarios where it is impractical to collect data with a complete rotation. Therefore, a direct comparison with the method proposed in this paper can be made. The CNFFFT convolution kernel, as described in the literature, operates on similar principles to the near-field-to-far-field transformation kernel presented in this paper. Both kernels play crucial roles in the FF transformation process.

In this experimental setup, the FF reference E-field and RCS are carefully configured for both HH and VV polarization modes, specifically to showcase the transformation outcomes of the CNFFFT approach. These results are then juxtaposed with the corrected signals derived from Section 3.2, as illustrated in Figures 10 and 11. The reference signal originating from the FF is represented by the red line, while the signal generated by CNFFFT is depicted by the blue line. Notably, the trends exhibited by these two lines are remarkably similar. On the other hand, the green line corresponds to the signal projected by the near-field-to-far-field transformation utilizing the amplitude extracted through the SSM method, as expounded in this paper. Upon careful examination of the figure, it becomes evident that the prediction accuracy achieved by this approach surpasses that of both the reference and CNFFFT signals.

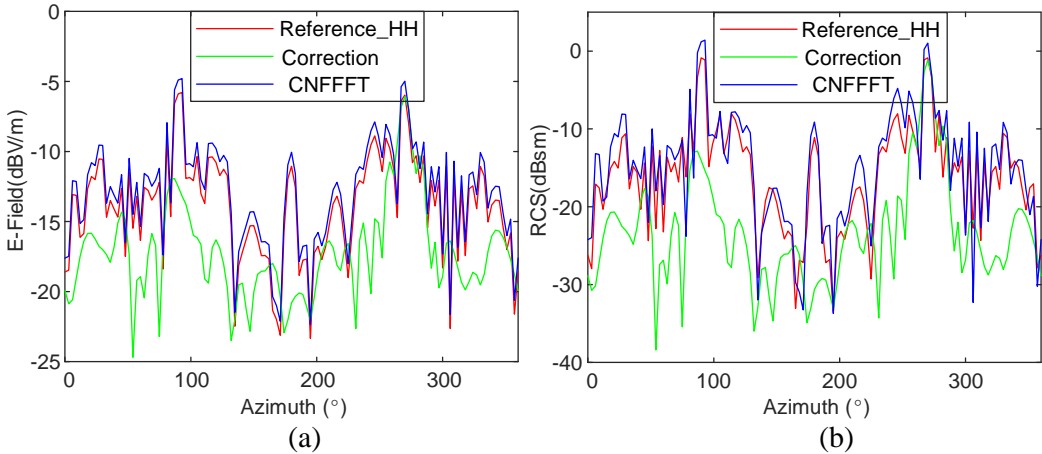

**Figure 10.** The NFFFT derived from the estimation of NF amplitude in HH mode compare with the reference of FF and CNFFFT: (**a**) E-Field; (**b**) RCS.

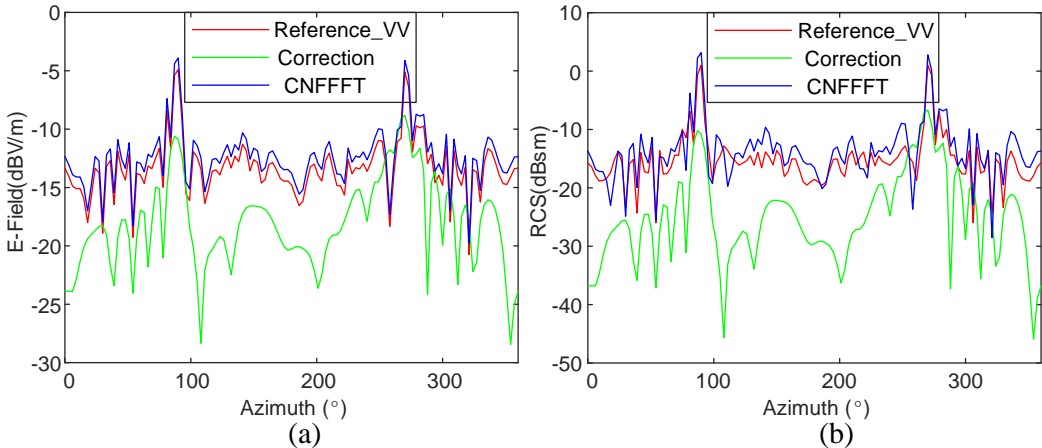

**Figure 11.** The NFFFT derived from the estimation of NF amplitude in VV mode compare with the reference of FF and CNFFFT: (**a**) E-Field; (**b**) RCS.

In summary, the proposed method outlined in this paper effectively enhances the precision of FF RCS prediction based on NF measurements.

*3.4. Calibration Discussion*

Real-world applications introduce a range of challenges that can influence the performance of the NFFFT method. These challenges encompass factors such as measurement noise, environmental interference, and the complexity of the measurement setup.

The pivotal role of calibration in the successful application of the NFFFT method is highlighted. Calibration ensures the proper characterization of the measurement system and setup, leading to precise transformation results. The significance of calibration and the potential impact of calibration errors on the final FF results are discussed. Additionally, different calibration techniques that can be employed to improve the transformation process's precision are explored.

In real-world scenarios, measurement artifacts may arise as a result of imperfections in the measurement equipment or environmental conditions. These artifacts can negatively affect the NFFFT, resulting in inaccuracies in the final results.

Drawing from the insights obtained through our research, practical recommendations are provided for researchers and practitioners to effectively apply the NFFFT method in real-world settings. These recommendations encompass best practices for measurement setup, calibration procedures, data preprocessing, and result validation.

## 4. Conclusions

The present study encompasses a comprehensive investigation into radar signal processing, with a specific focus on the estimation of the NF magnitude and its subsequent application in the inversion of the RCS from NFs to FFs. Theoretical values for the incident and scattered E-fields of the target, as well as the received field voltage, are derived and analyzed.

To bridge the gap between theoretical values and practical measurements, a novel approach utilizing the SSM is introduced to estimate the NF magnitude. This technique proves to be instrumental in accurately predicting the FF inversion, thereby mitigating discrepancies between theoretical expectations and actual measurements. The effectiveness of estimating the NF magnitude is demonstrated through a series of simulation experiments, which highlight the improvements in the accuracy of FF RCSs achieved through this approach.

The contributions of this work are threefold. Firstly, a comprehensive scattering model for NF targets is derived, providing a solid theoretical foundation for subsequent analyses. Secondly, the utilization of the SSM technique for the estimation of the NF magnitude is proposed, offering a robust and efficient method for amplitude-only estimation in the NF-to-FF transformation process. This contribution enhances the accuracy of FF inversion, leading to more reliable RCSs predictions. Lastly, a detailed derivation of the NF inversion for FF RCS is presented, further solidifying the theoretical framework and providing valuable insights into the underlying principles of the transformation process.

Overall, this study significantly advances our understanding of radar signal processing, particularly in the context of NF-to-FF transformation and RCS estimation. The derived theoretical models, coupled with the proposed SSM-based approach, contribute to the improvement of accuracy and reliability in predicting FF RCSs from NF measurements. The methods presented in this paper reveal certain aspects that demand further investigation in future research. One such aspect is the utilization of the discrete Fourier transform for (inverse)-Fourier transforms required in CNFFFT, which warrants additional discussion. These findings have implications for a wide range of applications in radar systems and signal processing, paving the way for further advancements in the field.

**Author Contributions:** Conceptualization, J.Z.; methodology, J.H.; software, J.H.; validation, J.H. and Y.D.; investigation, Y.D.; writing—original draft preparation, J.H.; writing—review and editing, J.H. and J.Z. All authors have read and agreed to the published version of the manuscript.

**Funding:** This work was funded in part by the National Natural Science Foundation of China (Grant No. 62271247, No. 61801212), in part by the Key Laboratory of Radar Imaging and Microwave Photonics (Nanjing Univ. Aeronaut. Astronaut.), the Ministry of Education, the Nanjing University of Aeronautics and Astronautics, Nanjing, 210016, China, in part by Innovation experiment competition cultivation (SAR image analysis system based on FPGA) (Nanjing Univ. Aeronaut. Astronaut.), in part by the Basic Ability Improvement Research Project of Young and Middle-aged College Teachers in Guangxi (2020KY570012).

**Institutional Review Board Statement:** Not applicable.

**Informed Consent Statement:** Not applicable.

**Data Availability Statement:** The study did not report any data.

**Conflicts of Interest:** The authors declare no conflict of interest.

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
