# Peer review of "Near-Field-to-Far-Field RCS Prediction Using Only Amplitude Estimation Technique Based on State Space Method"

_electronics, doi:10.3390/electronics12153371_

Round 1
Reviewer 1 Report
A brief summary:
Valuable approach for the transformation of the near-field (NF) to far-field (FF) RCS with adjusted amplitudes estimated using a state space approach for specific azimuth ranges.
General concept comments:
Do to dimensional constraints of e.g. anechoic chambers, it is only possible to measure the near-field the radar-cross-section (RCS) of a target under test, depending on the used frequency and size of the target itself. Therefore an approximation of the far-field RCS derived from the near-field is desired. The authors present an approach we firstly the measure near-field is corrected in amplitude and then transformed to the far-field. The authors also describe how to use the circular near-field-to-far-field transform (CNFFFT) to calculate the far-field on for a azimuth range instead of the full sphere.
The amplitude correction based on singular-value decomposition is a reasonable approach. Other possible approaches should at least be mentioned e.g. other eigen-based estimators or AR/ARMA models.
Further it would be nice to see a discussion about the problems resulting from the usage of the discrete Fourier transform for the (inverse)-Fourier-transforms needed for the CNFFFT.
Also a discussion about practical usage of the proposed approach should be including especially regarding problems that will arise e.g. calibration.
The result presentation is sufficient for the different distances and separate polarisations VV and HH.
Specific comments:
-There is a missing equation reference in line 255
-the legends in the result plots in Figure 10/11 could be right-oriented
The quality of the English language is sufficient.
